# Non-Invasive Biomarkers for Early Lung Cancer Detection

**DOI:** 10.3390/cancers14235782

**Published:** 2022-11-24

**Authors:** Harman Saman, Afsheen Raza, Kalyani Patil, Shahab Uddin, Tatjana Crnogorac-Jurcevic

**Affiliations:** 1Hamad Medical Corporation, Doha 3050, Qatar; 2Barts Cancer Institute, Queen Mary University of London, London EC1M 5PZ, UK; 3National Center for Cancer Care and Research, Hamad Medical Corporation, Doha 3050, Qatar; 4Translational Research Institute, Academic Health System, Hamad Medical Corporation, Doha 3050, Qatar; 5Dermatology Institute, Academic Health System, Hamad Medical Corporation, Doha 3050, Qatar; 6Laboratory of Animal Research Centre, Qatar University, Doha 2731, Qatar

**Keywords:** lung cancer, biomarkers, early detection, screening, microRNAs, circulating tumour DNA, DNA methylation markers, radiomics

## Abstract

**Simple Summary:**

Lung cancer remains the first cause of cancer worldwide. The main reason for this high rate of death from lung cancer is dissemination of the disease at the time of presentation to hospital due to late diagnosis. The aim of this article is to review and assess the effectiveness of different techniques, currently in use and that are upcoming, in early detection of lung cancer. We will present and evaluate the principles of developing such techniques and how to overcome challenges frequently facing researchers in the field of early lung cancer detection. Improvement in early detection would lower the rate of death and the societal burden of this often lethal condition.

**Abstract:**

Worldwide, lung cancer (LC) is the most common cause of cancer death, and any delay in the detection of new and relapsed disease serves as a major factor for a significant proportion of LC morbidity and mortality. Though invasive methods such as tissue biopsy are considered the gold standard for diagnosis and disease monitoring, they have several limitations. Therefore, there is an urgent need to identify and validate non-invasive biomarkers for the early diagnosis, prognosis, and treatment of lung cancer for improved patient management. Despite recent progress in the identification of non-invasive biomarkers, currently, there is a shortage of reliable and accessible biomarkers demonstrating high sensitivity and specificity for LC detection. In this review, we aim to cover the latest developments in the field, including the utility of biomarkers that are currently used in LC screening and diagnosis. We comment on their limitations and summarise the findings and developmental stages of potential molecular contenders such as microRNAs, circulating tumour DNA, and methylation markers. Furthermore, we summarise research challenges in the development of biomarkers used for screening purposes and the potential clinical applications of newly discovered biomarkers.

## 1. Introduction

Cancer is a leading cause of death worldwide, accounting for nearly 10 million deaths in 2020. Of all cancers, lung cancer (LC) is the second most common cancer type with 2.21 million new cases and 1.8 million deaths reported globally [1,2]. The overall 5-year survival rate remains low at 20%, which is mostly due to the advanced stage at the time of diagnosis [3], as in most cases early LC is asymptomatic. Patients with early disease often present with lung nodules or a mass revealed incidentally on a chest X-rays or CT scans. In advanced stages, LC can cause symptoms due to local tumour invasion, loco-regional spread, distant metastasis, and in some cases paraneoplastic syndromes. Common symptoms include cough (50–75%), haemoptysis (25–50%), shortness of breath (25%), and chest pain (20%) [4].

According to the World Health Organization (WHO), 85% of lung cancers are classified as non-small cell lung cancer (NSCLC) while the remaining are diagnosed as small cell lung cancer (SCLC). Histologically, lung cancer is classified as adenocarcinoma (AC), squamous cell carcinoma (SCC), large cell cancer, and other/unspecified subtypes (Figure 1).

In addition to these subtypes, SqCLC (about 20%) is classified into keratinizing, non-keratinizing, and basaloid types. Neuroendocrine tumours comprise four categories: SCLC, large cell carcinoma (LCC), typical carcinoid, and atypical carcinoid. SCLC (14% of all LCs) is further categorized into small cell carcinoma and combined small cell carcinoma. Both SCLC and large cell neuroendocrine carcinoma are high grade tumours. Carcinoid tumours are commonly located in central airways and further divided into two categories: typical carcinoid (intermediate grade) and atypical carcinoid (low grade). Adenosquamous carcinoma is rare and comprises 0.4–4% of LC cases [5]. The complexity of lung cancer is further increased by its highly heterogenous nature.

The most important risk factor for LC is smoking, which accounts for approximately 90% of all cases of LC [6,7]. Therefore, most research in LC screening focuses on the early detection of LC in current and former smokers [8], where if LC is caught early, this leads to the most benefit in terms of increasing life expectancy and increasing quality of life [8]. However, it is important to emphasise that LC screening is not an alternative to smoking cessation campaigns and that former smokers should continue to be encouraged to remain abstinent from smoking [9]. The discovery, development, and validation of different biomarkers for early detection of LC is likely to result in saving more lives in the not-too-distant future [10]. 

Common methods used for diagnosis and prediction of treatment response and disease progression include imaging and tissue biopsy. Both of these methodologies have their own limitations including cost, extensive patient preparation, risk of injury, invasiveness, exposure to radiation, and diagnostic bias due to heterogeneity. Therefore, it is imperative that non-invasive biomarkers are developed for screening/diagnosis, disease monitoring/prognostication, and prediction of response to treatments [11,12]. 

Numerous non-invasive biomarkers, such as DNA originating from tumour cells and circulating tumour cells (CTCs), proteins, lipids, RNAs, and microRNAs (miRNA), can be detected in bodily fluids such as plasma, serum, urine, saliva, ascites fluid, and CSF. Such cellular biomarkers are an area of extensive research mainly due to the ease of sampling and availability of validated sensitive technologies such as enzyme-linked immuno-sorbent assay (ELISA), polymerase chain reaction (PCR), next generation sequencing, colorimetric/electrochemical assays, and fluorescence methods [13,14,15,16]. However, due to the genomic instability and continuous evolution in lung cancer cells, a wide variation in expression of specified biomarkers is expected, and the ideal biomarker has not been found yet (Box 1).

Box 1Characteristics of ideal cancer biomarker.Safe and easy to collect and measure, with high validity and reliability.Readily available and cost-effective sources, such as urine.Not affected by important demographics such as gender or ethnicity.High SN to allow accurate detection of early stage of cancer.Requires minimum human input, especially if analysing a large amount of data, in order to reduce the impact on human resources.


Therefore, researchers are continuously searching for robust biomarkers with high sensitivity (SN) and specificity (SP) that can be used in clinical settings for early diagnosis. This will help in effective timely interventions and better patient management [17]. Following principles, summarised in Box 2, will enable and foster the progress in the biomarker field. 

Box 2Important principles in the field of development of non-invasive biomarkers for screening and early detection of LC.
A:A deeper knowledge of oncogenesis of LC is required to make sense of molecular and cellular complexities, and of the gene–environment interactions. B:Researchers and clinicians need to raise awareness and promote the significance of LC screening and early detection to the public governmental and non-governmental stakeholders with the view of funding research projects to develop single and integrated biomarkers to improve the efficacy of current lung cancer screening practices.C:There is a need for streamlining the processes involved in sample collection, utilising standard operating procedures, to overcome issues caused by heterogeneity in sample collection and analysis. D:The selection of candidate biomarkers and the test(s) used for their evaluation should be based on internationally agreed criteria. E:There is a need to establish a set of criteria to assess novel biomarkers in relation to their relevance and importance to clinical settings, taking into account cost effectiveness, reducing false-positive and false-negative rates, and satisfactory ratios of true- and false-positive results and their implications on service provision and logistics.


This review aims to provide information on several biomarkers that are being used as well as being investigated for the screening and early detection of lung cancer.

### 1.1. DNA Methylation in Sputum and Plasma for Early LC Detection

As epigenetic changes in LC are common, this offers several targets that can concurrently be probed [18]. LC genome analysis reports global hypomethylation that results in the destabilisation of DNA with the exception of CpG dense regions [19,20,21]. In NSCLC, epigenetic changes are associated with cigarette smoking and aggressive tumour behaviour, and as such these changes can be used for risk stratification and histological and molecular characterisation [22,23,24,25,26,27,28,29]. Non-invasive, sputum-based epigenetic testing for the detection of epigenetic changes/promoter DNA hypermethylation at early stages of tumorigenesis is well documented. Palmisano et al. showed that in sputum samples, collected 3 years prior to clinically detectable lung cancer, the hypermethylation of *MGMT* and/or *CDKN2A* genes could be effectively detected, indicating that epigenetic markers can indeed play a role in early cancer diagnosis [30]. This was validated in other studies as well [31,32,33]. Moreover, in a study of five participants, *RASSF1A* methylation, detected in sputum samples, correlated with the development of LCs within 12 to 14 months from the sputum test in three patients [34]. Similarly, a prospective study on 92 high risk individuals and a matched control group identified promoter methylation of 14 genes in the sputum that can be used for risk stratification. It was found that 6 of 14 genes correlated with a >50% increased LC risk. Furthermore, simultaneous methylation of three or more of these six genes correlated with 6.5-fold increased risk of LC [35]. These detected genes are involved in many important biological functions, such as cell cycle regulation (*p16* and *PAX5 β*), apoptosis (*DAPK* and *RASSF1A*), signal transduction (*GATA5*), and DNA repair (*MGMT*) [35,36,37,38,39].

The detection of DNA methylation in plasma, as a tool for screening and diagnostic purposes in LC, has also shown promise. Bearzatto et al. reported an increased frequency in *p16*^INK4A^ methylation in plasma samples of early-stage adenocarcinoma [40]. Similarly, methylations of *RASSF1A* and *CDKN2A* detected in blood samples were frequently identified in early-stage LC with a reported sensitivity of 22 to 66% and specificity of 57–100% [40,41,42]. Another study on 70 participants showed significant differences in the methylation pattern between LC and benign lung lesions. The participants who developed lung cancer showed methylation changes in four tumour suppressor genes, i.e., Kif1a, DCC, RARB, and NISCH. The differences were correlated with LC diagnosis, and it was observed that participants who were finally diagnosed with LC exhibited significant differences in methylation pattern [43]. Another, larger study on 360 participants showed similar results. The methylation status of *PTGER4* and *SHOX2* genes detected in the plasma of patients with indeterminate pulmonary nodules was distinct as compared to participants with benign lung nodules [44]. Therefore, integrating DNA methylation expression patterns (in plasma/sputum) as a screening tool in national LC screening programs is now needed to progress to novel algorithms for early LC detection. In lieu of this, Kang et al. developed a probabilistic method called Cancer Locator, based on cfDNA detected in blood samples. The study utilized data from a genome-wide DNA methylation profile and DNA methylation microarrays of solid tumour samples to train the model. The model was able to identify the histological type and the site of the tumour together with cancer load in NSCLC [45]. The study could not offer firm conclusions because of small sample numbers; however, the authors foresaw that when more paired samples (tumour sample and the matched adjacent non-tumour sample) become available, Cancer Locator could identify not just the existence but also the location of the tumour [45].

### 1.2. The Role of microRNAs in LC Detection

miRNAs are small non-coding RNAs of 18–25 nucleotides in length which are involved in the post-transcriptional regulation of gene expression [46,47]. They are found to be aberrantly expressed in many pathological conditions, including cancer, and can be detected in bodily fluids including urine, sputum, and blood, making them exciting biomarkers for cancer detection [48,49]. In 2002, their role in LC pathogenesis (proliferation of LC cells, invasion of basement membrane, and metastasis) was reported by Calin et al. [50]. Interestingly, based on the cellular context, miRNAs can act as tumour suppressors or oncogenes and even both [51,52]. Moreover, miRNAs preserve their stability throughout cancer progression from initiation to metastasis, because they are too small to degrade, and some miRNAs are further protected in exosomes. Hence, miRNAs are considered an appealing biomarker for cancer diagnosis and monitoring [53]. Several studies listed in Table 1 investigated miRNAs from different biofluid sources including sputum and serum/plasma for LC biomarker detection [54]. Previous studies of miRNAs in sputum showed that four miRNAs, miR-486, miR-21, miR200b, and miR-375, can differentiate lung adenocarcinoma patients from healthy individuals with an SN of 80.6% and SP of 91.7% [55]. Furthermore, a combination of miR205, miR-210, and miR-708 in sputum samples was able to discriminate squamous cell carcinoma patients from healthy controls with an SN of 73% and SP of 96% [56].

In an early study by Yanaihara et al., 12 microRNAs including miR-17-3p, miR-21, miR-106a, miR-146, miR-155, miR-191, miR-192, miR-203, miR-205, miR-210, miR-212, and miR-214 were identified as potential biomarkers of distinguishing cancers from benign lung tissues and as molecular markers, as they have different expressions in different malignant tissues [66]. Another study by Wozniak et al. [64] showed that a combination of 24 miRNAs was able to discriminate LC cases from healthy controls. The authors suggested that the overexpression of the above miRNAs in plasma can serve as a biomarker for the early detection of NSCLC and should be investigated further [64].

On the other hand, a study on circulating miRNA profile in the serum samples of 82 pre-operative LC patients, paired 10 days post-operative patients (before and after tumour removal), and 50 healthy participants showed increased expression of four miRNAs (miR-21, miR-205, miR-30d, miR-24) before surgery compared to after surgery and healthy participants. The researchers proposed that these four miRNAs have the potential to be used as biomarkers for post-operative disease relapse [61]. The same miRNAs were upregulated in the serum of early-stage LC patients in comparison to normal volunteers, suggesting that measuring their serum levels could potentially be extended for screening of high-risk subjects. As the serum levels of miR-21 and miR-24 were lower in post-operative compared to pre-operative patients, this feature should be investigated as a tool for monitoring disease recurrence in the post-operative setting [61]. In a similar study by Leidinger et al. [63], plasma miRNA levels were measured before surgery and at subsequent regular intervals up to 18 months post-surgery, with a reported significant correlation between miRNA expression level and time distance from surgery. The study indicated that, over time, the expression of specific miRNAs decreased. The post-surgery analysis of all miRNAs revealed a general reduction shortly after surgery and then a rise at disease progression. A network analysis showed that 12 miRNAs involved in controlling the regulation of 48 genes were deregulated in LC tissue and the level of miRNA expression change after surgery correlated with post-operative patients’ outcome and presence or absence of metastatic disease [63,67]. Therefore, due to the ability of miRNAs to change according to treatment dynamics, it is postulated that miRNA can be used for LC monitoring ad can provide prognostic information. A major issue with studies of miRNA as a tool for LC screening and early detection is the differences in protocols for sample collection and processing, combined with different assays for measuring miRNA expression employed by different studies, which result in variability in the obtained results. These differences in the methodologies should be taken into consideration as they potentially underlie the general lack of overlap in found miRNAs among the above-mentioned studies.

Another non-coding RNA type, circRNAs, which have a stable covalently closed circular structure and show a specific expression pattern in different tissues and cells, have also been implicated in LC growth and progression [7]. However, the exact mechanisms remain poorly understood and require more in-depth studies [8]. Using technologies such as RNA-seq and Ribo-Zero, thousands of circRNAs have been discovered ([7], and it is predicted that valid circRNA biomarkers for diagnosis, prognosis, and therapy in LC will increasingly be found. A better understanding of the exact role of circRNAs in the pathogenesis of LC will likely also lead to improvement of the detection of “clinically significant” circRNAs and understanding of the temporal relationship between such circRNAs and the development of preinvasive or early LC.

### 1.3. The Role of Circulating Tumour DNA (ctDNA) in LC

ctDNA (circulating tumour DNA) includes both encapsulated (in circulating vesicles) and non-encapsulated free DNA in the blood or other body fluids [68]. ctDNA escapes cancer cells via several mechanisms, namely apoptosis, necrosis, and secretion from extracellular vesicles as well as from CTCs [69,70]. Therefore, analysing ctDNA is a promising approach that could accelerate efforts for body fluid-based LC detection and overcome some of the challenges posed by invasive tissue biopsy, as summarised in Table 2.

An important feature of ctDNA is that it can be found in blood prior to clinical diagnosis [80]. Advances in technologies of DNA sequencing made it possible to detect cDNA before clinically evident LC [81]. However, a major challenge in using ctDNA is that most patients have ctDNA levels of less than 0.1% [82,83]. Nonetheless, new techniques have continuously been developed and tested to improve the detection of ctDNA in low concentrations in plasma. There is also evidence of a positive correlation between disease burden and the plasma concentration of ctDNA [81]. A study by Jacob et al. [80] used deep sequencing (CAPP-Seq) and improved protocol for the extraction of unique cfDNA fragments and the segment of cfDNA duplexes for sequencing of both strands [80]. The authors genotyped tumour tissue, analysed pre-treatment cfDNA in plasma and leukocyte DNA from 85 subjects diagnosed with stage I–III NSCLC using targeted deep sequencing of 255 frequently mutated genes in NSCLC, and reported that most somatic mutations in the cfDNA of LC patients and of risk-matched cohorts replicate clonal haematopoiesis and are not recurring. In contrast with mutation driving carcinogens, clonal haematopoiesis mutations are present on longer cfDNA fragments and do not show mutational marks that correlate with tobacco smoking. Incorporating these results with other tumour characteristics such as cell proliferation and lymphovascular invasion, the authors applied and prospectively validated a machine-learning-based method called “LC likelihood in plasma” (Lung-CLiP) [82]. Three control groups were used as a validation cohort: a low-risk group of 42 adult blood donors, a matched risk control group of 56 age, sex, and smoking status matched adults who had negative low-dose CT (LDCT) screening scans, and a third group comprising 48 risk-matched participants receiving LDCT screening recruited prospectively at a different centre. The study reported that Lung-CLiP successfully differentiates early-stage LC patients from risk-matched cohorts, with an overall 80% SP and SP of 63% in stage I, 69% in stage II, and 75% in stage III patients. Lung-CLiP performance was comparable to that of tumour-informed ctDNA detection, allowing tuning of assay specificity for the screening and early diagnosis of LC. The authors concluded that the potential of cfDNA for LC screening is strongly emerging and highlighted the significance of risk-matching LC cases and control groups in studies utilising cfDNA-based screening to account for hidden biases. The study proposed that Lung-CLiP could be used for high-risk subjects who decline LDCT due to concerns regarding false positives, limited access, and radiation exposure, by referring only individuals with positive Lung-CLiP test for further LDCT screening. One study suggested that this approach of integrating Lung-CLiP with LDCT could increase the number of lives saved in the US from LC from about 600 to approximately 12,000 by increasing the sensitivity of Lung-CLiP in detecting early lung cancer [84]. The study also noted a correlation between the pre-treatment levels of ctDNA and clinical outcomes, which might signify micro-metastasis even in early stages of LC, indicating the benefit of neoadjuvant and adjuvant systemic therapies. 

Another study reported a novel plasma-based assay for the diagnosis of early-stage LC exploiting high-throughput targeted DNA methylation sequencing of ctDNA [85]. The researchers established a methylation profile by high-throughput DNA bisulfite sequencing in tissue samples (nodule diameter of less than <3 cm) in order to separate cancerous from benign lesions. Subsequently, the methylation pattern was used to develop an assay for plasma sample classification using ctDNA. For example, in one study, the methylation profiling of 230 tissue samples was performed to study the cancer-specific methylation patterns which attained a 92.7% SN and 92.8% SP. These patterns were filtered using a training set of 66 plasma samples, and nine biomarkers were elected to construct a model for prediction of diagnosis. Sixty-six plasma samples were used for independent validation; this model achieved an SN of 79.5% and SP of 85.2% for discriminating patients with cancer (n = 39) from patients with benign pulmonary nodules (n = 27) [85]. 

One of the key shortcomings of molecular analysis by studying ctDNA is that it provides no information on histology; therefore, invasive biopsy will be required to make a histological diagnosis of LC. False-negative results from analysing ctDNA is a further important issue in the context of low tumour load or low rate of shedding of ctDNA to the systemic circulation [86]. Moreover, the precision of the data acquired by analysing ctDNA is affected by the location of the metastatic disease. A pooled analysis of EGFR-mutated NSCLC revealed that the detection rate of ctDNA EGFR mutation was considerably higher in patients with extrathoracic compared to intrathoracic lesions [79]. Furthermore, the false-positive results can be acquired using ctDNA as mentioned above (molecular alterations originated by clonal haematopoiesis rather than the tumour) [87]. Identification of unintended germline mutations during ctDNA evaluation that are not linked to the pathogenesis of LC is not an infrequent occurrence that mandates disclosure to the patient and referral for genetic counselling clinics [88]. For example, in the molecular analysis using ctDNA of 10,888 unselected patients with metastatic cancer (41% were lung malignancies), 1.4% were discovered to have possible hereditary cancer mutations in 11 genes [88]. Finally, technical aspects in relation to ctDNA specimen acquisition and handling can affect the quality of the data. Despite the many advantages of LBs compared to tissue biopsies, the SN and SP of detecting specific molecular changes in NSCLC from LB remain affected by technology, clinical trial methodologies, and logistics, which in turn affect the safe and effective integration of LB into clinical practice [89]. In a first published systematic review of 34 studies involving 1141 patients with NSCLC by Esagian et al., the positive percent agreement (PPA) in detecting common mutations using targeted NGS between LB and tissue biopsy was provided [90]. The authors stated that they used PPA rather than SN, SP, and PPV and NPV because NGS was not validated in all the studies they reviewed, and hence PPA was deemed more appropriate. The calculated PPA rates were 53.6% (45/84) for ALK, 53.9% (14/26) for BRAF, 56.5% (13/23) for ERBB2, 67.8% (428/631) for EGFR, 64.2% (122/190) for KRAS, 58.6% (17/29) for MET, 54.6% (12/22) for RET, and 53.3% (8/15) for ROS1. The above findings are consistent with other publications that concluded that the detection of specific mutations via NGS from LB is less sensitive compared to tissue biopsy [91,92].

### 1.4. Urine Cell-Free DNA (ucfDNA) in the Diagnosis of LC

Improvements in the knowledge and the technologies for the isolation and analysis of biomarkers from urine provide novel opportunities for the clinical applications of cancer urine biomarkers. The presence of biomarkers such as exfoliated bladder cancer cells, ctDNA, proteins, miRNAs, and exosomes in the urine have been investigated in the context of different primary cancers such as bladder, prostate, pancreas, and lung; the cost-effectiveness and convenience of use make urine biomarkers attractive choices for patients and physicians alike [93,94,95,96]. Using urine biomarkers for assessing treatment efficacy and resistance is a major advantage when compared to tissue biopsies and radiological imaging [97]. Furthermore, another advantage of urine biomarker analysis is that cfDNA extraction is technologically easier [97,98,99], when compared with plasma, as urine contains a lower concentration of interfering proteins [100]. The evidence for the reliability and sensitivity of the detection of gene mutations and DNA methylation in the urine is growing, especially as the technologies used are consistently undergoing refinement [101,102,103].

Methods associated with the extraction and classification of urinary constituents are multifarious and diverse and can vary from methods for protein and genomic profiling to microfluidic techniques [104]. In recent years, the detection of EGFR mutation and the subsequent mutation profile in patients with metastatic NSCLC who might be eligible to receive first and second lines of anti-EGFR tyrosine kinase inhibitors (TKIs) has grown rapidly. A study by Reckamp et al. showed that EGFR mutations (T790M, L858R, and exon 19 deletions) were successfully identified in the urine of NSCLC patients and the results were congruent with the EGFR mutation state identified through tissue biopsy [105]. A comparative study was reported by Ren et al., who measured the concentration of ucfDNA, using qPCR, in 55 LC patients and a cohort of 35 healthy participants [106]. The study reported that the concentration of ucfDNA is consistently higher in LC patients, especially with lymph node involvement, compared to the healthy cohort, suggesting that ucfDNA could potentially play a role in the early diagnosis of LC [106]. Another study compared the urine cell-free DNA (ucfDNA) of 55 NSCLC patients of different disease stages with 35 healthy volunteers by means of quantitative real-time PCR (qPCR) [107]. The study showed that concentrations of urinary cell-free DNA (ucfDNA) were considerably greater in individuals with stage III/IV than in those with stage I/II and the disease-free cohort. The receiver operating characteristic curves (ROCs) for distinguishing participants with stage III/IV from disease-free volunteers showed areas under the curve (AUCs) of 0.84 and 0.88, respectively. In another study [106], ucfDNA concentration and integrity indexes were explored as biomarkers for early LC detection. The cohort included 55 LC patients and 35 healthy participants. The study found that concentration and integrity indexes of ucfDNA were considerably higher in LC patients compared to the healthy individuals. Moreover, the ucfDNA integrity indexes in patients with metastasis to lymph nodes were significantly higher compared with patients without lymph node involvement, suggesting that ucfDNA could potentially play a role in the early diagnosis of LC [106].

### 1.5. RNA Airway and Nasal Signature

The approach of analysis of RNA acquired from airway samples centres on gene expression profiles of cancer-associated processes affecting the tracheobronchial tree [108]. A study identified a 23-gene biomarker panel from endobronchial brushings of patient who received bronchoscopy to investigate LC [109]. Consequently, two separate prospective cohorts showed an SN of 88% to 89% and an SP of 48% for such a gene-expression classifier. As biomarkers, these 23 genes were especially indicative of possible underlying cancer in patients with an intermediate (10–60%) pre-test risk of LC (91% negative predictive value, NPV). These results suggest that the NPV of a negative bronchoscopy could be improved if combined with the 23-gene panel, which could potentially circumvent the need for invasive lung biopsy by monitoring such patients with less invasive tests such as follow up CT scans [110]. In another study by the Aegis Study Team [111], the same concept of “field of injury” was used to investigate samples of nasal epithelial cells. The main advantage of this approach is bypassing the need for bronchoscopy. The investigators developed a 30-gene nasal expression panel for the detection of LC among smokers with suspected LC. This approach showed improvement in AUC, SN, and NPV if combined with clinical risk models. The study showed that combining clinical factors (age, smoking status, time since smoking cessation, tumour mass size) and the expression of the 30 genes from nasal cavity had a statistically significantly higher AUC (0.81; 95% confidence interval (CI) = 0.74 to 0.89, *p* = 0.01) and SN (0.91; 95% CI = 0.81 to 0.97, *p* = 0.03) than a clinical-factor only model [111]. 

### 1.6. Radiomics Signatures of Primary and Secondary Pulmonary Malignant Lesions

In the past decade, medical imaging has progressed from chiefly being a primary diagnostic tool to acquiring an important role in providing vital molecular data required for targeted based therapy through the adoption of advanced hardware, novel imaging agents, streamlined scanning protocols, and improvements in computational power [112]; thus, we will briefly discuss its role here. The technological advances have enabled the extraction and processing of a large amount of data from quantitative imaging, in a process called radiomics [112]. By utilising a characterisation algorithm, radiomics has the potential to unveil disease features that cannot be seen by the naked eye [113]. The process of radiomics involves obtaining sub-visual, yet quantitative, image characteristics in order to produce usable datasets from radiological films [114]. Radiomics data extracted from medical scans (e.g., CT and MRI scans) can be utilised to discover diagnostic, predictive, and prognostic data in patients with malignancy through comparison with objective response criteria such as overall and progression-free survival, and can also be combined with tumour molecular and genetic profile (genotype); the latter is referred to as radiogenomics [115]. The process of converting medical imaging into meaningful data typically involves four steps: (a) image acquisition and reconstruction, (b) region of interest segmentation, (c) feature extraction and quantification, and (d) building predictive and prognostic models, as illustrated in Figure 2.

As a new technology, radiomics is in its infancy; therefore, its clinical application is still limited. In the context of primary LC, a significant interest in using radiomics to predict the histological and molecular characteristics, response to treatment, and overall prognosis is raised. Several studies have been able to identify specific radiomics signatures that differentiate NSCLC from other benign and pre-invasive lesions, including the prediction of EGFR status and response to treatment with TKI [116,117,118,119,120,121,122,123], as well as histological subtype. For example, a retrospective study of 148 patients with histologically confirmed NSCLC found thirteen radiomics features that predict histological subtype (ALC vs. SqCLC) with AUCs of 0.819 and 0.824, respectively [124]. Several studies of radiomics signatures have reported features distinguishing benign from cancerous lung pathologies and are shown in Table 3. 

To conclude, radiomics offers a tangible opportunity for even wider use of medical imaging in oncology, especially in difficult to access lesions or lesions in patients in whom invasive lung biopsy could be detrimental.

## 2. Future Direction and Challenges 

### 2.1. Future Perspectives: Novel and Emerging Techniques

In order to improve the early detection of LC, future research should focus on examining the integration of currently used biomarkers with newly developed and emerging technologies. Examples of novel technologies that are currently under investigation are summarised below, with some of them being expected to enter clinical practice in the near future.

#### 2.1.1. Exhaled Biomarkers (EB), Volatiles, and Other Metabolites

Being completely non-invasive, EBs have been extensively studied to explore their potential for early LC detection and for the prediction of driver mutations. Robert et al. studied cells and DNA fragments isolated from EB and EB condensate (EBC) to detect a specific EGFR mutation (EGFR T790M) [131]. The authors concluded that detection of the EGFR T790M mutation from EBC can be a suitable and a non-invasive alternative to plasma samples. This is a welcome development as tests currently used to detect the EGFR T790M mutation in plasma have a low SN and SP [132]. 

In smokers, mutation in Kirsten rat sarcoma (KRAS) viral oncogene homolog is the most frequent driver mutation in NSCLC [133] as it occurs in 20 to 30% of all NSCLC patients [134]. KRAS mutations occur mostly (in 95%) at codons 12 and 13 [135]. Since 2009, the multi-institutional collaboration to study the genomic characteristics of lung adenocarcinomas, The Lung Cancer Mutation Consortium (LCMC), has analysed data from 1900 NSCLC patients and showed that 27% of lung adenocarcinomas harboured a KRAS mutation and about 30% of them harboured another oncogenic driver, most frequently STK11 and LKB1 mutation [136]. Patients with both KRAS and STK11 mutations had a considerably worse clinical outcome. With the emergence and therapeutic utility of mutant KRAS targeted TKIs, researchers are investigating alternatives to tissue biopsy for the accurate detection of KRAS mutations [133], especially as the role of using TKIs against actionable mutations has extended from metastatic to neoadjuvant and adjuvant settings [134]. A study of 19 patients by Kordiac et al. compared the status of KRAS mutation between EBC-DNA and cancer tissue and showed 46% concordance [133]. 

Another study investigated miRNA signature in NSCLC EBC in 21 age matched patients and an equal number of healthy controls. The concentration of 12 miRNAs was significantly altered in EBC from the LC patients, where a specific signature of miR-4507, miR-6777-5p, and miR-451a identified LC patients with high precision. Several targetable genes, namely CDKN2B, PTEN, TP53, BCL2, KRAS, and EGFR, were accurately detected in EBC. In a mouse model of lung adenocarcinoma development, driven by KrasG12D (the KrasLSL-G12D mouse), Rakhit et al. showed the potential utility of cfDNA and ctDNA to detect preinvasive lesions and the subsequent increase in the tumour size when followed by serial CT scans [135]. 

A pilot prospective study by Peled et al., of 72 patients with lung nodules [136], investigated EBs to profile volatile organic compounds by gas chromatography/mass spectrometry (GC-MS) combined with solid-phase microextraction and a chemical nanoarray. The authors concluded that breath analysis can distinguish benign from malignant nodules in a high-risk group for LC development. Similarly, a study of the response to systemic therapy in 39 patients with advanced LC showed promising results by assessing EB signature using gas chromatography/mass spectrometry and a nanomaterial-based array of sensors [137]. The study reported that GC-MS analysis identified three volatile organic compounds, namely alicyclic hydrocarbons and branched alkanes, as being considerably higher in the LC group compared to the control group. Moreover, the nanoarray could monitor variations in the concentration of these compounds in EBs in response to treatment, as well as resistance to therapy, therefore showing the potential to be used as a surrogate for response and resistance to treatment.

Several other metabolic biomarkers of LC are currently under investigation [138,139,140]. An advantage of metabolomics is that it allows direct functional information associated with the changes caused by LC and its tumour microenvironment [140]. Several studies focused on LC metabolomic biomarkers identified in blood, sputum, and EBC in order to pinpoint high-risk candidates for screening or to differentiate between benign and cancerous lung lesions [141,142,143]. For example, two important studies showed increased urinary excretion of creatine riboside and N-acetylneuraminic acid in patients with preclinical LC [144,145]. However, more robust studies involving large sample sizes are required to confirm these findings [145].

#### 2.1.2. Sputum Cell-Based Image Analysis

Enhanced cytology involves complex scan evaluation algorithms merged with artificial intelligence (AI) [146]. Meyer et al. developed a test to detect abnormal cells in the sputum of patients undergoing screening LC by LDCT [147]. The study reported an SN of 90% if at least 800 bronchial cells were obtainable for investigation; however, fewer cells were required when the clinical, molecular, or conventional sputum cytologic data were integrated with CT findings, and this provided a better diagnostic yield.

#### 2.1.3. Novel Ways of Utilising Genome Wide Association Studies (GWAS) for the Early Detection of LC

GWAS datasets represent a valuable source of information to evaluate the association between susceptibility genes and LC. Advancements in mathematical and statistical capabilities can integrate large numbers of single nucleotide polymorphisms (SNPs) with AI, risk models, and machine learning algorithms [148,149].

Several research groups are investigating the integration of LC genetic susceptibility genes into LC risk prediction models [150,151,152,153,154,155,156,157,158]. In the past 20 years, in excess of 1000 candidate gene association articles have been published on genetic predisposition to LC [159]. A metanalysis of 246 variants in 138 different genes showed 22 variants from 21 genes with significant correlation to predisposition to LC with significant epidemiological confirmation [159]. Nonetheless, thus far, the validity and reliability of LC susceptibility genes integrated risk prediction models are yet to be confirmed in order to be introduced into clinical practice [155,156,160].

#### 2.1.4. Transcriptomic, Proteomic, and Metabolic Signatures in Saliva 

Studies of other tumours such as breast, pancreas, and ovarian cancer have shown promising results in detecting cancer using transcriptomic, proteomic, and metabolomic signatures in saliva [161,162,163,164]. Proteomics is the large-scale study of proteomes, and a proteome is a set of proteins produced in both physiology and disease states [165], whereas metabolomics is the study of the entire set of metabolites present within an organism, cell, or tissue [164].

Using saliva as a body liquid has several advantages: it is easily accessible, amenable to self-sampling, and its analysis is potentially inexpensive. Nevertheless, a significant knowledge gap still exists that needs to be bridged before appropriate and accurate point-of-care instruments that can function as non-invasive diagnostic tools can be devised [166,167]. A review of the literature by Skallevold et al., which summarised transcriptomic, proteomic, and metabolomic signatures of LC in saliva, reported that such an approach has potential but requires more extensive validation [168]. The authors reviewed 27 articles in the timespan from 2011 to 31 December 2020 and reported that the published studies suffered from small sample sizes and large variation in the methodologies and technologies applied. Therefore, the study did not confirm the validity of using any of these -omics technologies in saliva for LC detection. 

Another study of 13 NSCLC patients to detect EGFR mutation using electric field-induced release and measurement (EFIRM) and droplet digital PCR (ddPCR) in plasma and saliva showed that EFIRM detected both EGFR mutations with a 100% SN in both plasma and saliva samples, whereas ddPCR detected EGFR mutations with sensitivities of 84.6% and 15.4%, respectively [169].

In conclusion, using omics technologies in saliva can offer complementary biological information in LC; however, more studies are required to improve the yield rate of detection of biomarkers through transcriptomics and proteomics approaches. Understanding the dynamic changes that occur in the salivary metabolome from preinvasive to advanced LC is another interesting topic that requires additional studies [170]. 

## 3. Challenges in the Development of LC Specific Biomarker

One challenge in the field of LC biomarker development has been the lack of standardisation and consistency in the process of sample collection, storage, and processing, which has likely adversely affected the results of systematic reviews and metanalyses, Box 3 [171]. This issue can be overcome by following a molecular approach, guided by the intended use and using standard operating procedures [171]. Another issue that affects the technical reliability of “omics” is the differences between laboratory procedures and the diversity in utilised platforms [172]. Heterogeneity within and between samples results in variations in measurements, caused by the very nature of cancer characterised by intra-tumour and inter-tumour heterogeneity which is difficult to control. Thus, real time profiling, follow-up biomarker panels, and tailored therapeutics need to be developed for an individual tumour, factoring in unique tumour characteristics.

Box 3Challenges in developing valid and reliable biomarkers in cancer.
(1)Absence of the “ideal” biomarker as a gold standard makes the validation of new cancer biomarkers for efficient cancer diagnosis, i.e., establishing clinical relevance and applicability, challenging.(2)Tumour evolution inevitably causes mutational diversity, resulting in inter-tumour and intra-tumour heterogeneity, which in turn cause variation in the quality and quantity of a biomarker in a specific primary tumour. (3)The complexity and dynamic range of a biomarker (particularly in plasma) make measuring its level reliably very difficult in the same patient or when compared among patients. (4)Low relative abundance of many disease-specific biomarkers often results in false-negative results, and therefore low SP. (5)Pre-analytical and analytical variables such as the method of sample collection, storage, transportation, and technologies used to measure the biomarker in question can lead to variable results and, therefore, adversely affect the validity and reliability of the biomarker.(6)Similar to the challenges facing new drug discovery, the development of novel biomarkers in cancer involves a complex, lengthy, and expensive pathway from bench to clinic. 


Large prospective cohort studies and databases such as the European Prospective Investigation into Cancer and Nutrition (EPIC) [173], medical records, and initiatives such as the UK Biobank [174] provide an unparalleled opportunity for continuing biomarker discovery and can provide solutions to some of the above challenges. International collaborative networks between researchers, such as the Early Detection Research Network at the US National Cancer Institute, offer the opportunity to exchange expertise, skills, and knowledge amongst researchers [175]. Analysis of pooled data from large cohorts such as from the UK [174], US [176], Denmark, and Germany [177] will address many statistical and logistical issues and concerns in the field of cancer biomarker development.

## 4. Conclusions

Currently, there is no single valid and reliable molecular biomarker in routine clinical practice that can be used for LC screening and/or early detection, despite the large number of candidate biomarkers and the advances made in the understanding of lung tumorigenesis, biomedical technologies, and applications of AI in data analysis. Therefore, the quest for discovering a valid and reliable early LC biomarker remains a priority in order to address this unmet clinical need.

Improving and standardising LC risk models will enable prioritising individuals at higher risk of LC for screening purposes. Validated risk models for LC screening would improve the efficiency and cost-effectiveness of national LC screening programs as well as provide the much-needed high quality clinical data for further investigation.

New technologies and improvement in processing digital data through AI and advanced mathematical models will allow utilising the vast amount of information obtained through CT and MRI scans that are currently carried out in large numbers on a daily basis. Radiomics and radiogenomics, through the integration of genomics data with the subvisual radiological data, have the potential to take LC screening, diagnosis, and treatment monitoring to the next level.

The growing need for predictive molecular and clinical biomarkers to evaluate screen-detected early-stage tumours requires researchers to focus on integrating promising candidate molecular and radiological biomarkers. This can be achieved by using modern mathematical and computer-based machine learning algorithms [178]. Moreover, sourcing high quality research-usable data requires a systematic collection of patient samples during screening programs. This can be achieved by introducing robust evidence-based data collection protocols that meet legislative requirements in relation to patients’ confidentiality and the sharing of information.

The use of ctDNA is an encouraging approach to systematically assess tumour molecular profiles in a minimally invasive way. Consequently, its role for early diagnosis and prognosis as well as disease and/or therapy monitoring in NSCLC and other cancers is expanding. In the clinical setting, the assessment of ctDNA is gradually becoming a common practice. However, a major hurdle is that, to date, only limited methods, assays, or platforms have been awarded approval by regulatory authorities [179]. Future studies should bring more sensitive methods for the detection of ctDNA at low levels from different clinical specimens. Furthermore, bioinformatic tools will be established, standardized, and validated in order to streamline clinical decision making.

Tumour heterogeneity and tumour evolution increase the complexity in developing precision medicine-driven biomarkers. Therefore, focusing on prospective studies that integrate both liquid biopsy and tissue biopsy (where possible) would be safer and more convenient in order to prevent the need for repeated invasive tissue biopsy, a common clinical scenario that can be faced with patients’ resistance and carry a high risk of complications. Table 2 summarises some important differences between tissue and liquid biopsy. This approach of integrating both methods of obtaining biopsy could offer the opportunity to study and understand the disease course and to monitor response to therapeutics. Discovering non-invasive biomarkers, therefore, could potentially be a more pragmatic method for evaluating tumour heterogeneity in clinical settings.

Although translating the latest data into clinical practice remains a challenge and is faced with significant hurdles, the era of the clinical use of reliable biomarkers is within reach, hopefully in the not-too-distant future. This aim can be achieved through stringent application of appropriate experimental strategies and trial execution. Rigorous standardisation of sample collection, storage, processing, and analysis will assure the validity and reliability of upcoming LC biomarker candidates.

## Figures and Tables

**Figure 1 cancers-14-05782-f001:**
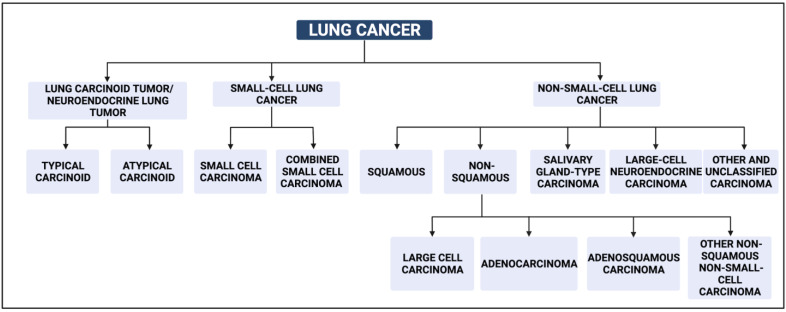
WHO 2015 classification of LC.

**Figure 2 cancers-14-05782-f002:**
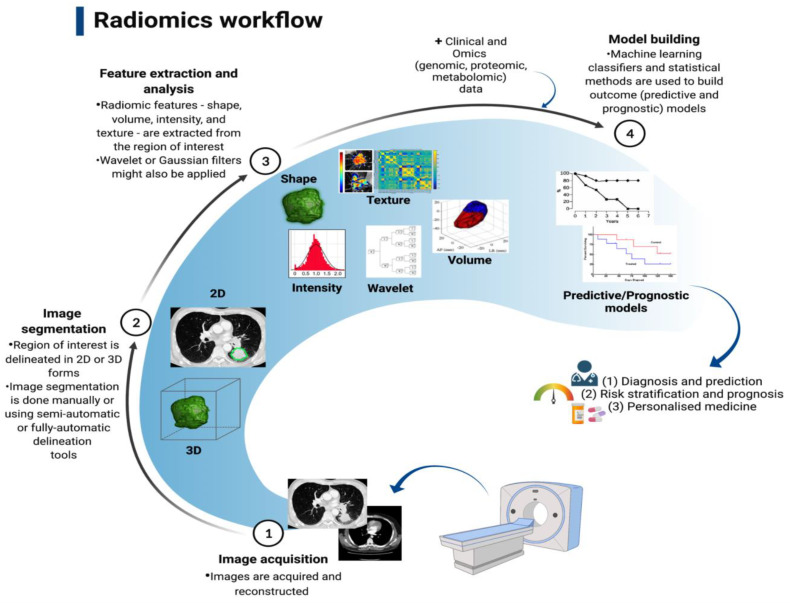
Radiomics workflow that involved four stages, Lambin et al. [113].

**Table 1 cancers-14-05782-t001:** Summary of studies showing miRNAs from different biofluid sources as potential biomarkers in LC.

Body Fluid	miRNA	Biomarker Utility	Sensitivity (SN) and Specificity (SP)	Reference
**Sputum**	miR-486, miR-21, miR-200b, miR-375	Potential use for diagnosis of NSCLC (ALC)	SN: 80.6% SP: 91.7%	[55]
**Sputum**	miR-205, miR-210, miR-708	Potential use for diagnosis of NSCLC (SqCLC cell)	SN: 73% SP: 96%	[56]
**Circulating exosomes**	miR-17-3p, miR-21, miR-106a, miR-146, miR-155, miR191, miR-192, miR203, miR-205, miR-210, miR-212, miR-214	Screening of lung ALC	SN: 65% SP: 89%	[57]
**Blood**	miR-222-3p, miR-22-3p, miR-93-5p	Prognostic marker of NSCLC (ALC)	SN: 65% SP: 88%	[58]
**Blood**	miR-26a-5p, miR-126-3p, miR-130b-3p, miR-205-5p, miR-21-5p	Prognostic marker of NSCLC (SqCLC)	SN: 62% SP: 89%	[58]
**Serum**	miR-23b, miR-221, miR-148b, miR-423-3p	LC diagnosis	SN: 79% SP: 92%	[59]
**Serum**	miR-145, miR-20a, miR-21	NSCLC	SN: 71% SP: 88%	[60]
**Serum**	miR-21, miR-24	LC recurrence	SN: 74% SP: 93%	[61]
**Serum**	miR-21, miR-205, miR-30d, miR-24	LC diagnosis	SN: 69% SP: 87%	[61]
**Plasma**	miR-126, miR-145, miR-210, miR-205-5p	LC diagnosis	SN: 72% SP: 95%	[62]
**Plasma**	miR-34a, let-7c	LC recurrence	Not available	[63]
**Plasma**	miR-122, miR-182, miR-193a-5p, miR200c, miR-203, miR-218, miR-155, let-7b, miR-411, miR-450b-5p, miR-485-3p, miR-519a, miR-642, miR517b, miR-520f, miR-206, miR-566, miR-661, miR-340, miR-1243, miR-720, miR-543, miR-1267	Early NSCLC diagnosis	SN: 81% SP: 89%	[64]
**Plasma**	miR-155, miR-197, miR-182	Early LC detection	SN: 83% SP: 88%	[65]

**Table 2 cancers-14-05782-t002:** Important differences between LB (analysis of ctDNA) and tissue biopsy.

	Analysis of ctDNA	Tissue Biopsy
**Accessibility and convenience**	Blood (and other body fluids)-based tests. This makes it more accessible for sample collection and acceptable by patients.	Invasive and often requires exposure to radiation.
**Factors affecting SN and SP**	ctDNA levels are also influenced by disease burden and many other factors such as tumour location, vascularity, and cellular turnover [71,72].	Accessibility of the tumour, patient’s fitness and personal preference, tumour heterogeneity.
**The effect of tumour type on the detection of ctDNA in blood and other body fluids**	Tumours in the central nervous system or those with mucinous features (such as prostate and thyroid) frequently show low or undetectable ctDNA levels [73,74].	Not applicable.
**Cost effectiveness**	More cost-effective and time-efficient than tissue biopsy [75].	The cost could soar, especially if biopsy from difficult location requires operation, e.g., surgical brain biopsy.
**Histological diagnosis**	Provides no information regarding histology.	Is required to make a histological diagnosis.
**Monitoring disease progression and response to treatment**	Has an established role in treatment response monitoring or the early detection of relapse [76,77].	Not always possible or practical due to its invasive nature.
**As a screening biomarker**	Can be used for population-based screening [78].	Not possible or practical due to its invasive nature.
**Detection of minimal residual disease (MRD)**	The role in detecting MRD after curative treatment is growing.	Not applicable.
**The effect of the location of metastasis on the accuracy of the result**	The SN of analysis of ctDNA to detect EGFR mutation in the setting of NSCLC is greater in intrathoracic compared to extra-thoracic located tumours [79].	Not applicable.

**Table 3 cancers-14-05782-t003:** Summary table showing studies of radiomics signatures to distinguish benign from cancerous lung pathologies.

Name of Study	Number of CT Scans	Radiomics Feature	Statistical Tool Used to Assess Performance
Ardila et al. [125]	Data extracted from NLST: 6630 benign 86 malignant Independent validation set: 1112 benign 27 malignant	1024 radiomics features were assessed and validated by expert radiologists.	AUC of training dataset: 0.944 AUC of validation dataset: 0.955
Chen, et al. [126]	33 benign 42 malignant	Support vector machine (SVM) was used as the classifier. 76 out of 750 characteristics were appreciably distinctive between benign and malignant nodules. Accuracy for the selected 4-feature signature (SFS) was the maximum.	SFS: Accuracy: 84% SN: 92.85% SP: 72.73%
Choi et al. [127]	72 pulmonary nodules, 31 benign and 41 malignant	103 radiomic signatures were tested.	Accuracy: 84.6% AUC: 0.89
Delzell et al. [128]	90 benign 110 malignant	416 radiomic signatures. Combinations of the 6 feature selection methods and 12 classifiers were examined by applying a 10-fold repeated cross-validation framework with 5 repeats.	AUC: 0.747 SN: 61.6% SP: 72.9%
Hawkins et al. [129]	Data extracted from NLST: 328 benign 170 malignant	219 radiomic signature with best model finding 23 stable signatures. J48, JRIP (RIPPER), Naïve Bayes, support vector machines (SVMs), and random forest(s) classifiers tested.	Accuracy: 80% AUC: 0.83
Peikert et al. [130]	Data extracted from NLST: 318 benign 408 malignant	LASSO logistic regression model implemented. 8 out of 57 radiomic signatures utilised.	AUC: 0.939

## Data Availability

Not applicable.

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
