# Peer review of "Non-Invasive Biomarkers for Early Lung Cancer Detection"

_cancers, 2022, doi:10.3390/cancers14235782_

Round 1

Reviewer 1 Report

In this article the Harman Saman and collegues provide an overview of the different diagnostic strategies for lung cancer. The authors describe classification of lung cancers followed by diving deeper into the different diagnostic methods. The subtopics convered include DNA methylation testing, noncoding RNAs microRNAs, ctDNA, urine cell free, RNA in airways. Next the authors discuss radiomics in lung diagnostics followed by analysis of studies on exhaled biomarkers in sputum etc.

The text is supported by 3 tables 2 comprehensive figures and 162 references.

Minor comments:

The authors may well include a brief section on KRAS mutation and co-mutations in lung cancer diagnosis and prognosis

There could be some forward looking statements on the utility of circRNAs in LC diagnostics

Legends to figures can be enhanced

Minor typos were found throughout the manuscript that should be checked and corrected.

Author Response

Dear Sir/Madam ,

please  see the attached cover letter containing the responses to the reviewers' comments and the updated manuscript to reflect these responses. 

KR

Harman Saman

Reviewer 2 Report

I fully agree with the authors that there is an urgent need to identify and validate non-invasive biomarkers for early diagnosis, prognosis and treatment of lung cancer to improve patient management. Despite recent progress in the identification of non-invasive biomarkers, there is currently a lack of reliable and available biomarkers that demonstrate high sensitivity and specificity for detecting lung cancer. In this review, the authors have tried to cover the latest developments in this area, including the usefulness of biomarkers currently used for screening and diagnosing lung cancer. 1. Figure 1 is of very poor quality, it is not readable, it needs to be redone. 2. In table 1, I would like to see data on sensitivity / specificity. 3. In clause 1.3. there is not enough summary table on sensitivity / specificity according to different studies for greater clarity. 4. The promising methods of item 2 do not reflect all modern developments in the diagnosis of lung cancer, for example, using saliva https://doi.org/10.1007/s00018-012-1027-0, https://doi.org/10.3390 /diagnostics10040186 etc. or methods of infrared spectroscopy of blood serum https://doi.org/10.1016/j.saa.2013.11.049 etc. Nothing is said about the proteome and metabolome, which is also being actively studied. It seems to me that a description of all new promising developments will greatly decorate the manuscript.

Author Response

Dear Sir/Madam,

please  find the attached cover letter containing the responses to the reviews' comments and the updated manuscript to reflect these responses.

KR

Harman Saman

Round 2

Reviewer 2 Report

I have no more remarks/comments on the article. I believe that in its present form the manuscript can be recommended for publication.